# KGGen: Extracting Knowledge Graphs from Plain Text with Language Models

**Belinda Mo**[*1], **Kyssen Yu**[*2], **Joshua Kazdan**[*1], **Joan Cabezas**, **Proud Mpala**[1], **Lisa Yu**[2], **Chris Cundy**[3], **Charilaos Kanatsoulis**[1], **Sanmi Koyejo**[1]

[1]Stanford University    [2]University of Toronto    [3] FAR AI

## Abstract

Recent interest in building foundation models for knowledge graphs has highlighted a fundamental challenge: knowledge graph data is scarce. The best-known knowledge graphs are primarily human-labeled, created by pattern-matching, or extracted using early NLP techniques. While human-generated knowledge graphs are in short supply, automatically extracted ones are of questionable quality. We present KGGen, a novel text-to-knowledge-graph generator that uses language models to extract high-quality graphs from plain text with a novel entity resolution approach that clusters related entities, significantly reducing the sparsity problem that plagues existing extractors. Unlike other KG generators, KGGen clusters and de-duplicates related entities to reduce sparsity in extracted KGs. Along with KGGen, we release Measure of Information in Nodes and Edges (MINE), the first benchmark to test an extractor's ability to produce a useful KG from plain text. We benchmark our new tool against leading existing generators such as Microsoft's GraphRAG; we achieve comparable retrieval accuracy on the generated graphs and better information retention. Moreover, our graphs exhibit more concise and generalizable entities and relations. Our code is open-sourced at https://github.com/stair-lab/kg-gen/.

## 1 Introduction

Knowledge graph (KG) applications and graph-based Retrieval-Augmented Generation (RAG) systems are increasingly bottlenecked by the scarcity and incompleteness of available KGs. KGs consist of a set of subject-predicate-object triples, and have become a fundamental data structure for information retrieval [Schneider, 1973]. Most real-world KGs, including Wikidata [contributors, 2024], DBpedia [Lehmann et al., 2015], and YAGO [Suchanek et al., 2007], are far from complete, with many missing relations between entities [Shenoy et al., 2021]. The lack of domain-specific and verified graph data poses a serious challenge for downstream tasks such as KG embeddings, graph RAG, and synthetic graph training data.

Several synthetic plain-text-to-KG extractors have been proposed to address KG scarcity, most prominently including OpenIE [Angeli et al., 2015] and Microsoft's GraphRAG [Larson and Truitt, 2024]. Both OpenIE and GraphRAG extract entities and relations directly from text, but they lack effective mechanisms for entity resolution and relation normalization. This leads to graphs with nearly as many unique relation types as edges, resulting in sparse, disconnected knowledge representations that limit their utility for downstream tasks. To solve this problem, we propose KGGen, a text-to-knowledge-graph generator that leverages language models (LMs) and an algorithm for entity and edge resolution to extract high-quality, dense KGs from text. First, KGGen uses an LM-based extractor to read unstructured text and predict subject-predicate-object triples to capture entities and relations; after extracting the triples, it applies a novel, iterative clustering algorithm to refine the raw graph. Inspired by crowd-sourcing strategies for entity resolution [Wang et al., 2012], KGGen

---

[*]Equal Contribution.

39th Conference on Neural Information Processing Systems (NeurIPS 2025).

identifies nodes that refer to the same underlying entities, and consolidates edges that have equivalent meanings.

The nascent field of plain-text-to-knowledge graph extraction currently lacks benchmarks to measure the fidelity of KG generation from text. To close this gap, we provide two new benchmarks: the first captures information retention from short texts; the second, based on WikiQA, measures knowledge retrieval capabilities for graphs generated from multi-million token, web-based knowledge databases. On these benchmarks, KGGen performs comparably to GraphRAG. However, KGGen exhibits far better scaling properties with respect to information compression and graph sparsity as the plain-text database length increases.

To summarize our contributions:

1. We introduce KGGen, an open-source package that uses LMs to extract high-quality KGs from plain text. Our package is available as a Python library.

2. We develop benchmarks to drive improvements in plain-text-to-knowledge-graph extraction, and measure KGGen's performance on these benchmarks.

3. We show that KGGen exhibits improved scaling with respect to the size of the text source relative to past methods.

## 2   Related Work

Interest in automated methods to produce structured text to store ontologies dates back to at least 2001 when large volumes of plain text began to flood the fledgling internet [Maedche and Staab, 2001]. KG extraction from unstructured text has seen significant advances through rule-based and LM-powered approaches in the last 15 years. Early work [Suchanek et al., 2007] used hard-coded rules to develop YAGO, a KG extracted from Wikipedia containing over five million facts, and rules-based extraction still has appeal for those producing KGs in multi-modal domains today [Norabid and Fauzi, 2022, Oramas et al., 2015]. With the development of modern natural language processing, hard-coded rules generally ceded to more advanced approaches based on neural networks. For instance, OpenIE [Angeli et al., 2015] provides a two-tiered extraction system: first, self-contained clauses are identified by a classifier; then, [Angeli et al., 2015] run natural logic inference to extract the most representative entities and relations from the identified clauses. Stanford KBP [Angeli et al., 2013] presents another seminal early approach to using deep networks for entity extraction.

As early as 2015, some hypothesized that extracting KGs would go hand-in-hand with developing better language models [Domeniconi et al., 2015]. More recently, evidence has emerged that transformer-based architectures can identify complex relationships between entities, leading to a wave of transformer-based KG extraction techniques, which range from fully automatic [Qiao et al., 2022, Arsenyan et al., 2023, Zhang and Soh, 2024] to human-assisted [Kommineni et al., 2024]. Our contribution to the extraction literature is to build KGs conducive to embedding algorithms such as TransE and TransR [Bordes et al., 2013, Lin et al., 2015]. We observed that when one extracts KGs from plaintext, the nodes and relations are often so specific that they are unique. This causes the estimation of embeddings to be under-specified. We develop a method for automatic KG extraction from plain text that clusters similar nodes and edges to prevent this under-specification. This leads to a KG with better connectivity and more functional nodes and edges.

Evaluating the quality of knowledge graphs is important to ensure usefulness and reliability in downstream applications. Early evaluation methods focused primarily on directly assessing aspects such as completeness and connectivity or using rule-based statistical methods, while recent approaches emphasize usability in downstream applications and incorporation of semantic coherence[Xue and Zou, 2023].

In the late 2000s, research focused on assessing the correctness and consistency of KGs. The evaluations relied on expert annotations by selecting random facts from the generated KG and then calculating the accuracy of those facts. [Suchanek et al., 2007] This proved to be laborious and prone to errors. This led to accuracy approximation methods like KGEval [Ojha and Talukdar, 2017] and Two-State Weight Clustering Sampling(TWCS) [Gao et al., 2018], which employed sampling methods with statistical guarantees as well as use less annotation labor. As the KGs became larger and more diverse, particularly with the rise of automated extraction techniques from web data, this

generated more pressure on annotators, leading to methods like Monte-Carlo search being used for the interactive annotation of triples [Qi et al., 2022]. Furthermore, because accuracy alone did not fully capture the complexity of the knowledge graph, more evaluation metrics like completeness were used to characterize the quality of knowledge graphs. [Issa et al., 2021].

In recent years, the evaluation of knowledge graphs (KGs) has increasingly focused on their role in downstream AI applications, such as augmenting language models [Schneider et al., 2022] and recommendation systems [He et al., 2020]. As a result, semantic coherence and usability have become key criteria for assessing the quality of extracted knowledge graphs.

Two notable approaches to KG evaluation are the LP-Measure and the triple trustworthiness measurement (KGTtm) model. LP-Measure assesses the quality of a KG through link prediction tasks, eliminating the need for human labor or a gold standard [Zhu et al., 2023]. This method evaluates KGs based on their consistency and redundancy by removing a portion of the graph and testing whether the removed triples can be recovered through link prediction tools. Empirical evidence suggests that LP-Measure can effectively distinguish between "good" and "bad" KGs. The KGTtm model, on the other hand, evaluates the coherence of triples within a knowledge graph Jia et al. [2019]. Based on these evaluation methods, frameworks like Knowledge Graph Evaluation via Downstream Tasks(KGrEaT) and DiffQ(differential testing) emerged. KGrEaT provides a comprehensive assessment of KGs by evaluating their performance on downstream tasks such as classification, clustering, and recommendation [Heist et al., 2023] rather than focusing solely on correctness or completeness. In contrast, DiffQ uses embedding models to evaluate the KG's quality and assign a DiffQ Score, resulting in improved KG quality assessment Tan et al. [2024].

## 3 Existing Methods

Before describing KGGen, we explain two popular existing methods for extracting KGs from plain text, which will serve as a basis for comparison throughout the rest of this paper.

### 3.1 OpenIE

Open Information Extraction (OpenIE) was implemented by Stanford CoreNLP based on Angeli et al. [2015]. It first generates a "dependency parse" for each sentence using the Stanford CoreNLP pipeline. A trained classifier then traverses each edge in the dependency parse, deciding whether to create, continue, or stop processing a clause. These decisions split complex sentences into shorter, self-contained clauses. From these clauses, the system produces (*subject, relation, object*) tuples, each accompanied by a confidence score. Because OpenIE does not require its input text to have a specific structure, OpenIE can handle text in any format.

### 3.2 GraphRAG

GraphRAG, developed by Microsoft in 2024, integrates graph-based knowledge retrieval with language models (LMs) [Larson and Truitt, 2024]. As a first step, GraphRAG provides functionality for generating KGs from plain text, which serve as its database for retrieval. In this process, GraphRAG creates a graph by prompting LMs to extract node-entities and relationships between these entities. Throughout this extraction, few-shot prompting provides the LM with examples of desireable extractions. GraphRAG aggregates well-connected nodes into communities and generates a summary for each community. The final graph consists of the nodes and their relationships along with communities their summaries.

## 4 KGGen: Knowledge Graphs From Plain Text

Unlike most previous methods of LLM-based KG extraction, we rely on a multi-stage approach (1) extract entity and relations from each source text using an LLM, (2) aggregate graphs across sources, and (3) iteratively resolve duplicate entities and edges using a hybrid of LLM and traditional informational retrieval methods.

We impose strong constraints on the LLM via prompting to prevent it from incorrectly grouping together entities or edges that are similar in meaning but not actually the same - for example, conflating

"Type 1 diabetes" and "Type 2 diabetes," "hypertension" and "stress," or "MRI" and "CT scan.". We introduce multiple passes through our extracted edges and edges to resolve similar entities and consolidate the number of edge types. Entity and edge resolution prevents the formation of sparse KGs, which may produce meaningless KG embeddings under standard algorithms such as TransE.

Our extraction method involves several steps, which we outline below. The exact prompts for each step can be found in Appendix A, and the process is illustrated in Figure 1.

### 4.1 Entity and Relation Extraction

The first stage takes unstructured text as input and produces an initial knowledge graph as extracted triples. We use Google's Gemini 2.0 Flash as the language model to provide structured output via DSPy signatures. The first step takes in source text and extracts a list of entities. Given the source text and entities list, the second step outputs a list of subject-predicate-object relations. Each step corresponds to a DSPy signature that specifies instructions for the LLM to follow in its docstring. We find this 2-step approach works better to ensure consistency between entities.

### 4.2 Aggregation

After extracting triples from each source text, we collect all the unique entities and edges across all source graphs and combine them into a single graph. All entities and edges are normalized to be in lowercase letters only. The aggregation step reduces redundancy in the KG. Note that the aggregation step does not require an LLM.

### 4.3 Entity and Edge Resolution

After extraction and aggregation, we typically have a raw graph containing duplicate or synonymous entities and possibly redundant edges. The resolution stage is a key innovation in our KG extraction methodology that merges nodes and edges representing the same real-world entity or concept. Our resolution process employs a two-stage approach combining embedding-based clustering with LLM-based de-duplication to efficiently handle large knowledge graphs. The approach is applied to both entity and edge items separately:

First, all items in the graph are clustered. We get the semantic embeddings of every item using S-BERT and cluster using k-means into clusters of 128 items.

1. For each item in a cluster, we retrieve the top-k most semantically similar items, where k=16, using a fused BM25 and semantic embedding approach.

2. Then, the LLM is prompted to identify exact duplicates from this set, considering variations in tense, plurality, case, abbreviations, and shorthand forms.

3. For each set of duplicates, the LLM selects a canonical representative that best captures the shared meaning, similar to aliases that Wikidata uses. Cluster maps track which entities belong to which alias.

4. The item and its duplicates are removed from the cluster and steps 1-3 repeat until no items remain in the cluster.

This approach enables effective de-duplication even for very large knowledge graphs by processing semantic clusters in parallel. When processing our largest 20M chunk dataset, this method successfully consolidated entities like "Olympic Winter Games", "Winter Olympics", and "winter Olympic games" into a single canonical representation.

## 5 Benchmarks for Extraction Performance

Although a handful of existing methods attempt to extract KGs from plain text, it is difficult to measure progress on this task due to the lack of existing benchmarks. As a remedy, we produce the Measure of Information in Nodes and Edges (MINE), the first benchmark that directly measures a knowledge-graph extractor's ability to capture and distill a body of text into a KG. MINE contains two tasks: the first assesses how well a KG extractor captures the information in short, page-length text;

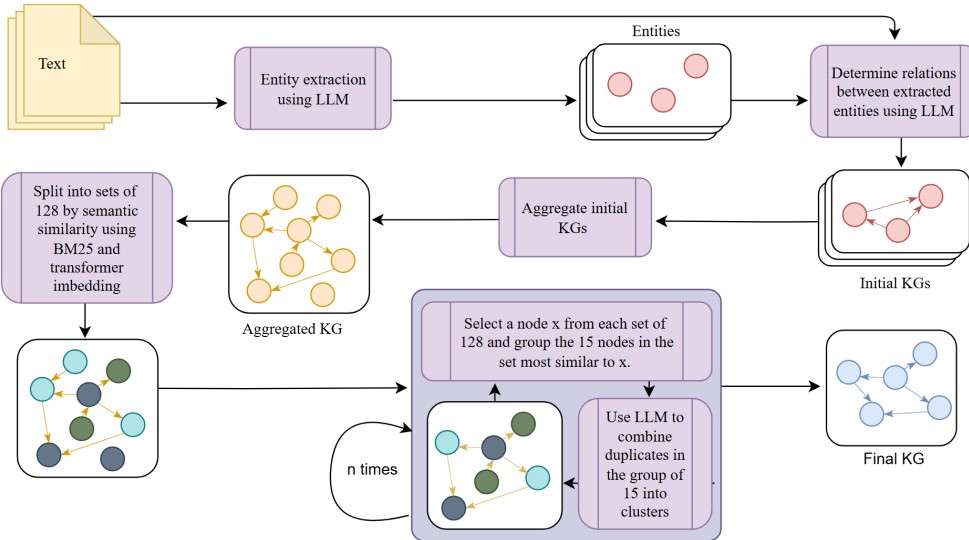

Figure 1: KGGen extraction method

the second measures downstream RAG performance on multi-million token datasets. We call these tasks MINE-1 and MINE-2 respectively. MINE-1 ensures that a KG extractor accurately represents the source text, and MINE-2 gauges the practicality of the knowledge graphs for realistic applications.

## 5.1 MINE-1: Knowledge Retention

MINE-1 approximates the fraction of information a KG extractor is able to capture from an article without relying on downstream tasks, which can obscure whether performance gains stem from the KG extractor itself or from aspects of the extraction process.

MINE-1 consists of 100 articles, each accompanied by 15 facts that are known to be present in the article. The dataset has the following characteristics: articles have a mean length of 592 words (std. 85 words, range: 440-976 words) and cover diverse topics including Arts, Culture & Society (24 articles), Science (27 articles), Technology (19 articles), Psychology/Human Experience (18 articles), and History & Civilization (17 articles). Articles are generated by an LLM to ensure balanced coverage across these domains. For each article, MINE-1 generates a corresponding KG using the extractor being evaluated.

To assess the quality of these KGs, we extract 15 facts from each article using an LLM-based extraction prompt found in Appendix B. We manually verify that the 15 facts are accurate and contained in the article. To measure performance on MINE-1, the KG extractor first extracts a KG from each article. Then, MINE-1 contains a process to verify whether each fact can be recovered from the corresponding KG.

Verification occurs via a semantic query process: both the 15 facts and all KG nodes are embedded using the all-MiniLM-L6-v2 model from SentenceTransformers. For each fact, the verifier retrieves the top-k most semantically similar nodes in the KG, then expands the result to include all nodes within two relations of those top-k nodes, reflecting the fact that KG's are often used for multi-hop reasoning tasks. The subgraph induced by these nodes is passed to an LLM, which is prompted to output a binary score: 1 if the fact can be inferred from the retrieved nodes and relations alone, and 0 otherwise. The prompt is detailed in Appendix B. The final MINE-1 score for a KG extractor is the percentage of the 15 facts scored as 1, averaged across all 100 articles. While LLM-based evaluation introduces potential biases, we validated its reliability by manually scoring 60 randomly selected fact-KG pairs and comparing them to LLM judgments, achieving 90.2% agreement and a correlation of 0.80. The full evaluation pipeline is illustrated in Figure 2.

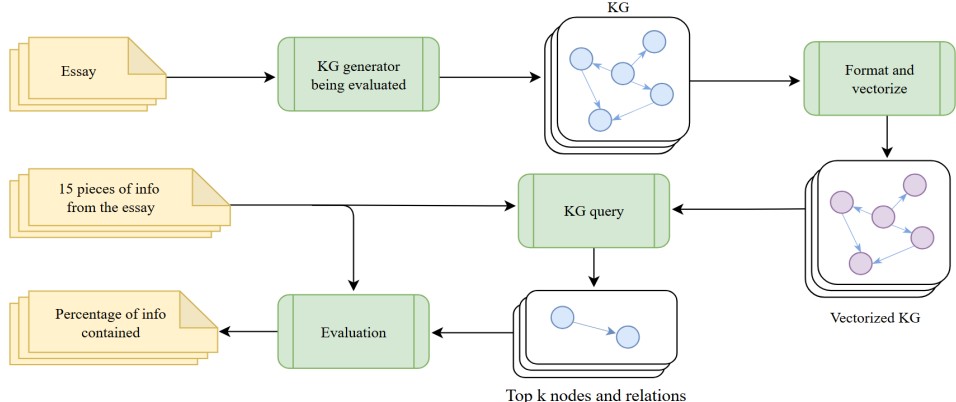

Figure 2: Evaluation process used in MINE-1

## 5.2   MINE-2: KG-Assisted RAG Description

The RAG evaluation is based on the WikiQA dataset Yang et al. [2015], which contains 20,400 questions based on 1,995 Wikipedia articles. Using the method under evaluation, we construct a single KG that aggregates information from all articles referenced in WikiQA. For each question in the dataset, we retrieve the top 10 most relevant triples by embedding both the question and all KG triples using the all-MiniLM-L6-v2 model from SentenceTransformers. We then compute the cosine similarity between the question and each triple, alongside a relevance score produced by BM25. The final similarity score is obtained by combining BM25 relevance score and the cosine similarity score, weighted equally. The 10 triples with the highest combined scores are selected, and we further expand this set by appending 10 additional triples that fall within two hops of the nodes in the top 10 triples to enable multi-hop reasoning.

Since each relation is linked to a source text chunk during generation by KGGen and GraphRAG, we provide the full set of 20 retrieved triples, their associated text chunks, and the original question to an LM, which synthesizes an answer based on these inputs. The complete prompt used can be found in Appendix B. Finally, the LM responses are evaluated using LLM-as-a-Judge to determine whether they contain the correct answer to the question. The prompt used for this final verification step is also included in Appendix B. OpenIE is excluded from this comparison, as it cannot produce KGs that link relations to the original text chunks.

## 6   Results

We use MINE to benchmark KGGen against leading existing methods of plain-text-to-KG extraction: OpenIE Angeli et al. [2015] and GraphRAG Larson and Truitt [2024]. After providing this quantitative comparison of extraction fidelity, we present qualitative results that demonstrate the advantages of KGGen over past methods.

### 6.1   Evaluations on MINE-1

Figure 3 displays accuracies from KGGen, OpenIE, and GraphRAG on MINE. Note that KGGen outperforms competing methods. Figure 4 shows an example query from MINE-1 and relevant relations extracted by KGGen, OpenIE, and GraphRAG.

#### 6.1.1   Generalization Across Language Models

To evaluate KGGen's robustness across different foundation models, we tested its performance on MINE-1 using multiple state-of-the-art LLMs. Table 1 shows that KGGen maintains strong performance across different models, with Claude Sonnet 3.5 achieving the highest score of 73%.

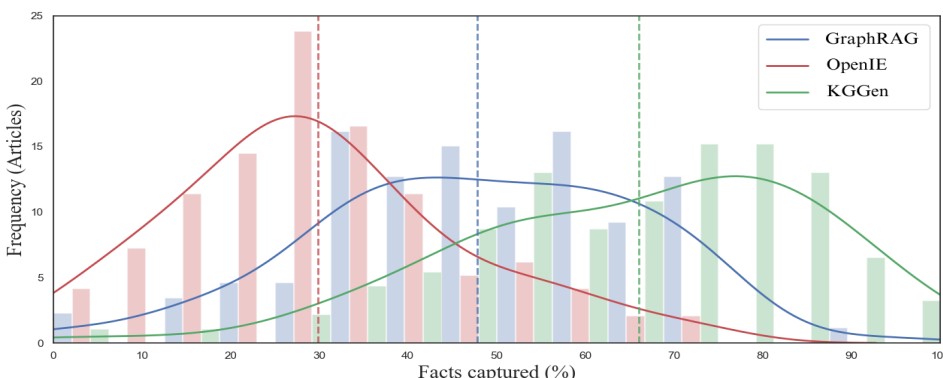

Figure 3: Distribution of MINE-1 scores across 100 articles for GraphRAG, OpenIE, and KGGen. Dotted vertical lines show average performance. KGGen scored $66.07\%$ on average, significantly outperforming GraphRag $47.80\%$ and OpenIE $29.84\%$.

| Fact being queried for: | "Decentralization provides users with more control over their funds in cryptocurrencies." | |
|---|---|---|
| Extractor | Sample of relevant triples queried from KG | Result |
| KGGen | (**cryptocurrencies**, enhance, **security**) (**cryptocurrencies**, are, **decentralized**) (**cryptocurrencies**, provide control over, **funds**) (**cryptocurrencies**, enhance, **privacy**) (**cryptocurrencies**, operate on, **peer-to-peer network**)(**cryptocurrencies**, revolutionizing, **transactions**) (**blockchain**, ensures, **transparency**) | 1 |
| GraphRAG | (**CRYPTOCURRENCIES**, Cryptocurrencies are having a profound impact on the financial world by introducing new ways of thinking about money and finance, **FINANCIAL WORLD**) (**BLOCKCHAIN**, Cryptocurrencies operate using blockchain technology which provides a secure and transparent way to record transactions, **CRYPTOCURRENCIES**) | 0 |
| OpenIE | (**cryptocurrencies**, allowing transactions to occur between users, **without need for intermediaries**) (**cryptocurrencies**, allowing, **for transactions to occur directly**) (**Cryptocurrencies**, have taken financial world in, **storm**) (**Blockchain**, is, **ledger technology**) (**Blockchain**, is distributed, **ensures**) | 0 |

Figure 4: An example query from the MINE-1 benchmark, along with relevant relations in the KGs extracted by KGGen, GraphRAG, and OpenIE. Note that the relation triples extracted by KGGen contain the fact being queried for, whereas the KGs extracted by GraphRAG and OpenIE do not. The relation types extracted by KGGen are more concise and generalize more easily than those from GraphRAG and OpenIE. The full article that these relations were extracted from can be found in Appendix C.

Table 1: Performance comparison of KGGen

(a) KGGen performance on MINE-1 across different language models

| Model | MINE-1 Score (%) |
|---|---|
| Claude Sonnet 3.5 | 73 |
| GPT-4o | 66 |
| Gemini 2.0 Flash | 44 |

(b) Validity of extracted triples across different methods

| Method | Valid Triples (%) |
|---|---|
| KGGen | 98/100 (98%) |
| GraphRAG | 0/100 (0%) |
| OpenIE | 55/100 (55%) |

KGGen's extraction methodology generalizes well across different foundation models. Although Claude Sonnet 3.5 achieves the highest score of 73%, all tested models maintain reasonable extraction quality, making KGGen adaptable to users' preferred LLM providers.

## 6.2 Evaluations on MINE-2: RAG performance

Figure 5 shows comparable performance between KGGen and GraphRag on MINE-2.

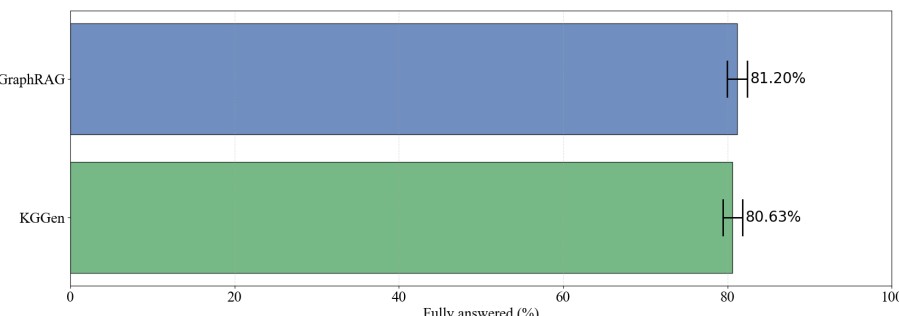

Figure 5: Comparison between KGGen and GraphRag on MINE-2. The two methods perform comparably.

## 6.3 Evaluation on Human-Annotated Data: SemEval-2010

To evaluate KGGen's extraction quality against human-annotated ground truth, we conducted experiments using the SemEval-2010 Task 8 dataset. We randomly selected 100 sentences from the dataset, each containing two manually labeled target entities. After removing entity markup tags to ensure unbiased extraction, we applied KGGen to extract entities and relationships from each sentence.

Our evaluation focused on entity extraction accuracy, as the dataset's relation labels consist of only 5 broad categorical types rather than specific semantic relations. We assessed whether both human-annotated target entities appeared in KGGen's extracted entities, allowing for more specific entity descriptions (e.g., "Eurasia exhibition" instead of just "exhibition") that still referred to the same object.

Results show that KGGen successfully captured both target entities in 96% of cases (96/100). The method consistently extracted more detailed entity descriptions compared to human annotations, often identifying additional relevant entities in longer sentences. For example, given the sentence "The ambitious Eurasia exhibition arose from an idea by Achille Bonito Oliva," with target entities 'exhibition' and 'idea', KGGen extracted ['Eurasia exhibition', 'idea', 'Achille Bonito Oliva'], providing more specific and complete entity identification.

## 6.4 Qualitative Results

We first evaluated the fundamental quality of extracted knowledge graphs by examining whether the extracted triples conform to the basic definition of a knowledge graph: subject-predicate-object triples where subjects and objects are entities (nodes) and predicates are relationships (edges). We randomly selected 100 triples from each method and manually evaluated their validity. The results are shown in Table 1.

Despite GraphRAG's comparable performance on downstream tasks, it does not extract structures that closely resemble traditional knowledge graphs, which is a major strength of KGGen. As seen in Figures 6b and 6e, GraphRAG often extracts very few nodes and connections for an entire article. This sparsity results in the omission of critical relationships and information. For compression, Figure 6a and 6d illustrate sections of the KGs generated by KGGen for the same articles. Figure 6c illustrates one of many issues in OpenIE's KGs. Firstly, most node types are hyperspecific, incoherent phrases. Many of these nodes are redundant near-copies of each other, adding unnecessary complexity to the graph. Additionally, as seen in 6f OpenIE primarily uses pattern matching to identify entities, and frequently produces generic nodes such as "it" and "are". Due to their frequency, these nodes, which contain no useful information, often end up as some of the most well-connected nodes in the graph. Consequently, unrelated concepts end up being just two hops apart, linked by paths through nodes like "it" or "are". By contrast, KGGen consistently generates KGs that are informative and coherent, effectively capturing critical relationships and information from the articles.

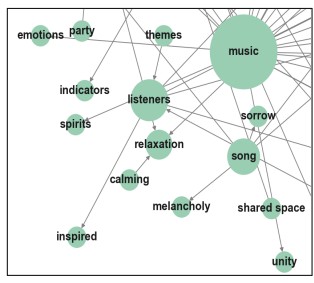

(a) Section of KG generated by KGGen on "How Music Influences Mood"

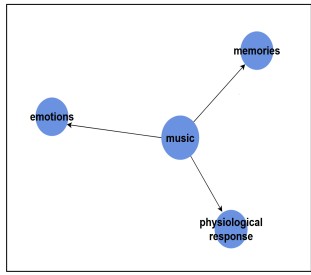

(b) Full KG generated by GraphRAG on "How Music Influences Mood"

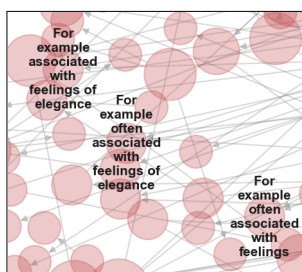

(c) Section of KG generated by OpenIE, on "How Music Influences Mood", with most node labels omitted for readability.

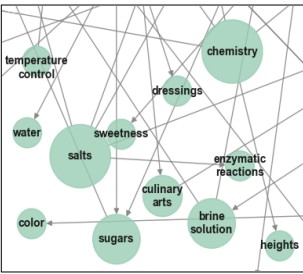

(d) Section of KG generated by KGGen on "The Chemistry of Cooking"

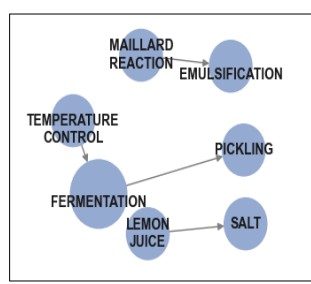

(e) Full KG generated by GraphRAG on "The Chemistry of Cooking"

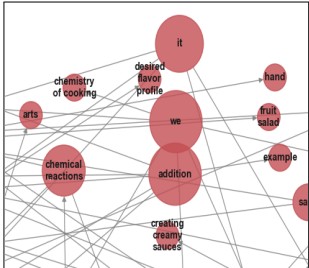

(f) Section of KG generated by OpenIE on "The Chemistry of Cooking"

Figure 6: Visual comparison of KGs generated using KGGen, GraphRAG, and OpenIE. Results show that KGGen discovers more informative nodes to estimate a richer graph compared to GraphRAG, and collapses synonyms to discover a more informative graph than OpenIE.

## 6.5 A Note on Scaling

A major motivation for the creation of KGGen was to produce graphs where edge types are generalizable, and used more than once when the corpus grows large. To test the re-usability of our relations, we generate three knowledge graphs from text of different sizes: 10 000 characters, 100 000 characters, and 1 000 000 characters and plot the number of edges divided by the number of unique relations. As one can see from Figure 7, KGGen reuses each relation-type an average of 10 times, and the average number of occurrences of each relation-type increases with the size of the corpus. By contrast, GraphRAG reuses each relation type an average of 2 times regardless of the size of the graph. This suggests that the relations in GraphRAG do not generalize as the corpus grows.

## 6.6 Efficiency and Cost Analysis

To evaluate the practical applicability of KGGen, we analyzed its computational efficiency and cost on a large-scale extraction task. We extracted a KG from the novel *The Name of the Wind* by Patrick Rothfuss, processing text corpora of increasing sizes. Table 2 demonstrates KGGen's de-duplication effectiveness across different scales.

Table 2: KGGen scaling characteristics showing entity and relation de-duplication

| Corpus Size (chars) | Pre-Entities | Post-Entities | Entity De-dup Ratio | Pre-Relations | Post-Relations | Relation De-dup Ratio | Pre-Edges | Post-Edges | Edge De-dup Ratio |
|---|---|---|---|---|---|---|---|---|---|
| 100 | 1 | 1 | 1.000 | 1 | 1 | 1.000 | 1 | 1 | 1.000 |
| 1,000 | 20 | 18 | 0.900 | 12 | 12 | 1.000 | 9 | 9 | 1.000 |
| 10,000 | 90 | 78 | 0.867 | 78 | 75 | 0.962 | 62 | 57 | 0.919 |
| 100,000 | 727 | 604 | 0.831 | 926 | 924 | 0.998 | 498 | 424 | 0.851 |
| 1,000,000 | 4,602 | 3,573 | 0.776 | 8,137 | 8,094 | 0.995 | 3,180 | 2,448 | 0.770 |

The de-duplication ratios improve with scale, demonstrating the effectiveness of our clustering algorithm. For the complete novel (1M characters), KGGen achieved a 22.4% reduction in entities

and 23% reduction in edges through intelligent consolidation. This analysis highlights the contribution of each module: the extraction phase (Steps 1-2) captures comprehensive information from text, while the resolution phase (Step 3) significantly reduces redundancy without information loss. Table 3 presents the computational cost breakdown for processing the entire novel:

Table 3: KGGen cost and throughput analysis for 1M character corpus

| Step | Prompt Tokens | Completion Tokens | Total Tokens | Time (s) | Throughput (tokens/s) | Cost ($) |
|---|---|---|---|---|---|---|
| KG Extraction (Steps 1–2) | 1.59M | 0.63M | 2.22M | 273 | 8,139 | 0.46 |
| Entity/Edge Resolution (Step 3) | 2.93M | 0.22M | 3.15M | 279 | 11,304 | 0.38 |
| **Total** | 4.52M | 0.85M | 5.37M | 551 | 9,739 | 0.84 |

For comparison, we evaluated GraphRAG on the same corpus: results can be found in Table 4.

Table 4: GraphRAG scaling characteristics on the same corpus

| Corpus Size (chars) | Entities | Relations | Edge Types | Time (s) |
|---|---|---|---|---|
| 100 | 2 | 1 | 1 | 1.89 |
| 1,000 | 4 | 3 | 3 | 3.01 |
| 10,000 | 16 | 20 | 20 | 29.71 |
| 100,000 | 80 | 100 | 99 | 205.12 |
| 1,000,000 | 514 | 981 | 966 | 2,079.17 |

While GraphRAG is faster on short corpora, its execution time scales superlinearly. The complete extraction time comparison reveals a significant difference: GraphRAG requires 2,319 seconds for the extraction phase alone on the 1M character corpus, compared to KGGen's total processing time of 551 seconds (including both extraction and resolution). Additionally, GraphRAG produces nearly as many relation types (966) as edges (981), indicating minimal relation reuse and poor generalization compared to KGGen's efficient consolidation.

## 7    Broader Impact and Community Adoption

Our work produces a KG-from-plain-text extractor that helps to solve the KG-scarcity problem. Improved knowledge-graph extractors could lead to the prevalence of structured text, which can help improve factuality and reliability of information retrieval systems. Our open-source implementation has already enjoyed widespread community adoption. **The package has received over 700 Github stars and has been downloaded over 12,000 times since its release.**

## 8    Limitations and Future Work

Although KGGen holds many advantages over past extraction methods, its graphs still exhibit problems, like over or under de-duplication of entities and relations. Further research into entity resolution could improve the quality of our KGs. Additionally, our benchmarks measure corpora of up to 5M tokens, which does not reflect the size of web-scale text that would be necessary to produce a KG foundation model. Future expansions of our benchmark could focus on larger corpora to better measure the practicality of different extraction techniques.

Domain-specific knowledge extraction presents additional challenges. Fields like medicine and finance require specialized domain knowledge that general-purpose LLMs may lack, potentially limiting extraction quality compared to human experts. While MINE-2 demonstrates KGGen's capability across diverse domains, incorporating domain-specific ontologies could improve extraction precision. Future work could explore adaptive ontology integration to balance structure with completeness.

## Acknowledgments and Disclosure of Funding

We acknowledge support from NSF grant numbers DGE-1656518, NSF 2046795 and 2205329, the MacArthur Foundation, Stanford HAI, and Google Incorporated.

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

# A    Prompts for KG Extraction

This section provides the exact prompts used to extract KG's from the text.

The initial KG was extracted using the following two prompts passed as DSPy signature descriptions.

**Prompt for extracting entities:**  `Extract key entities from the source text. Extracted entities are subjects or objects. This is for an extraction task, please be thorough and accurate to the reference text.`

**Prompt for extracting relations:** `Extract subject-predicate-object triples from the source text. Subject and object must be from entities list. Entities provided were previously extracted from the same source text. This is for an extraction task, please be thorough, accurate, and faithful to the reference text.`

After extracting the entities and relations from each unit of text, we begin the de-duplication process, which is performed using the following prompts.

**Prompt for entity or edge resolution:**
`Find duplicate {item_type} for the item and an alias that best represents the duplicates. Duplicates are those that are the same in meaning, such as with variation in tense, plural form, stem form, case, abbreviation, shorthand. Return an empty list if there are none.`

# B    Prompts for MINE

This section provides the LLM prompts used by MINE to evaluate KGs.

**Prompt for extracting a fact from article:** `Extract 15 basic, single pieces of information from the following text that describe how one object relates to another. Present the pieces of info in short sentences and DO NOT include info not directly present in the text. Your output should be of the form [ "info1", "info2" ,..., "info15" ]. "Make sure the strings are valid Python strings."`

**Prompt for evaluating if a fact is contained in the query result:**
`ROLE: "You are an evaluator that checks if the correct answer can be deduced from the information in the context.`
`TASK: Determine whether the context contains the information stated in the correct answer.`
`Respond with "1" if yes, and "0" if no. Do not provide any explanation, just the number.`

**Prompt for RAG response:**
```
Use the following knowledge graph triples and text evidence to
answer the question.
Triples:  {triples_text}
Text Evidence:  {text_block}
Question:  {query} Answer:
```

**Prompt for Evaluating containment of WikiQA answer:**
```
You are a fact-checking assistant
Question:  question
Expected answer:  expected
Model's response:  response
Does the model's response contain the information in the expected
answer?
Respond with one word:  Yes or No.
```

## C   Example Article from MINE

This section provides the article that the example fact is from.

**Title:** The Rise of Cryptocurrencies

**Content:** Cryptocurrencies have taken the financial world by storm in recent years, revolutionizing the way we think about money and transactions. From the creation of Bitcoin in 2009 by an anonymous individual or group known as Satoshi Nakamoto, to the thousands of altcoins that have since emerged, cryptocurrencies have become a significant player in the global economy.One of the key factors contributing to the rise of cryptocurrencies is the decentralized nature of these digital assets. Unlike traditional fiat currencies that are controlled by governments and central banks, cryptocurrencies operate on a peer-to-peer network, allowing for transactions to occur directly between users without the need for intermediaries. This decentralization not only provides users with more control over their funds but also enhances security and privacy.Another driving force behind the popularity of cryptocurrencies is the technology that underpins them - blockchain. Blockchain is a distributed ledger technology that ensures the transparency and immutability of transactions on the network. Each transaction is recorded in a block and linked to the previous block, forming a chain of blocks that cannot be altered once validated by the network. This technology has been instrumental in building trust and confidence in cryptocurrencies, as it eliminates the need for a trusted third party to oversee transactions. The concept of decentralization and blockchain technology has also paved the way for various applications beyond just digital currencies. Smart contracts, for example, are self-executing contracts with the terms of the agreement directly written into code. These contracts automatically enforce and execute themselves when predefined conditions are met, eliminating the need for intermediaries and streamlining processes in various industries. Cryptocurrencies have also gained traction due to their potential for financial inclusion. In many parts of the world, traditional banking services are inaccessible or too costly for a significant portion of the population. Cryptocurrencies offer a way for individuals to access financial services, such as transferring money and making payments, without the need for a traditional bank account. This has the potential to empower individuals in underserved communities and drive economic growth. The volatile nature of cryptocurrencies has attracted both investors seeking high returns and speculators looking to capitalize on price fluctuations. The rapid appreciation of certain cryptocurrencies, such as Bitcoin, has led to a surge in interest from retail and institutional investors alike. While this volatility presents opportunities for profit, it also poses risks, as prices can fluctuate dramatically in a short period. Regulation has been a contentious issue in the cryptocurrency space, with governments and regulatory bodies grappling with how to oversee this emerging asset class.

Some countries have embraced cryptocurrencies and blockchain
technology, recognizing their potential for innovation and
economic growth.  Others have taken a more cautious approach,
citing concerns about money laundering, tax evasion, and consumer
protection.  Despite the challenges and uncertainties surrounding
cryptocurrencies, their rise has been undeniable.  As more
individuals and businesses adopt digital currencies for transactions
and investments, the landscape of finance is evolving rapidly.  The
future of cryptocurrencies remains uncertain, but their impact on
the financial world is already profound.  In conclusion, the rise
of cryptocurrencies can be attributed to their decentralized nature,
blockchain technology, financial inclusion potential, investment
opportunities, and regulatory challenges.  As these digital assets
continue to gain acceptance and adoption, they are reshaping the
way we think about money and finance.  Whether cryptocurrencies will
become mainstream or remain on the fringes of the financial system
remains to be seen, but their impact is undeniable and will likely
continue to unfold in the years to come.

## D  Additional Graphs

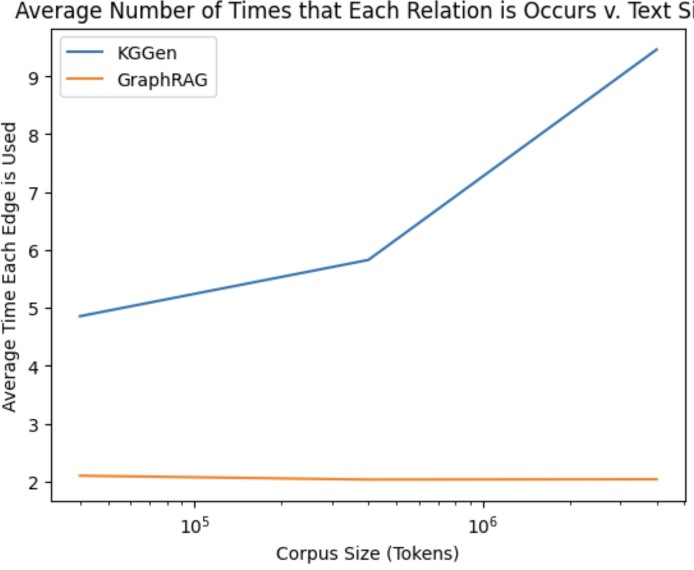

Figure 7: As graphs increase in size, KGGen tends to reuse each unique relation type more frequently, while GraphRAG maintains a consistent average usage of about 2 instances per relation type regardless of graph size.

