# OpenReview forum: "KGGen: Extracting Knowledge Graphs from Plain Text with Language Models"
_NeurIPS.cc/2025/Conference — NeurIPS 2025 poster_

### Official Review · Reviewer_S7Xx · 2025-06-17

**Clarity:** 3
**Significance:** 3
**Originality:** 3
**Rating:** 4
**Confidence:** 2

**Summary:**

KGGen introduces a novel language model-based method for extracting knowledge graphs (KGs) from plain text, addressing KG sparsity through entity/relation resolution. Key contributions include: (1) A three-stage pipeline (entity/relation extraction → aggregation → iterative resolution) that clusters synonymous entities/edges using hybrid embedding+LLM deduplication; (2) MINE, the first benchmark for text-to-KG extraction, with tasks for knowledge retention (MINE-1) and downstream RAG performance (MINE-2); (3) Empirical validation showing KGGen outperforms GraphRAG by 18% and OpenIE by 36% on MINE-1, achieves comparable RAG results to GraphRAG on MINE-2, and demonstrates superior relation reuse during scaling. The tool is released as a Python library.

**Questions:**

1. Could the resolution stage incorporate domain-specific constraints (e.g., medical ontologies) to reduce under-clustering of specialized terms?

2. What is the throughput (tokens/second) of KGGen on 1M-token corpora, and how does cost scale vs. GraphRAG?

3. In MINE-1, why use an LLM for fact verification instead of human annotation? Could model biases affect scores?

4. The authors claims KGGen has already enjoyed widespread community adoption. KGGen has received over 450 Github stars and has been downloaded as a python module over 11,000 times since its release in January. Thus, KGGen has already made an impact in the community that studies knowledge graphs. Does this violate the double-blind policy?

**Ethical Concerns:**

["NO or VERY MINOR ethics concerns only"]

**Final Justification:**

Thanks for the authors' response. I have read it and decided to maintain my positive score.

**Limitations:**

1. Generalizability: Evaluation primarily on Wikipedia-style articles; performance on conversational or technical text is unknown.

2. Black-box dependencies: Entity merging relies on Gemini without error analysis (e.g., confusion rates for ambiguous entities).

**Quality:**

3

**Strengths And Weaknesses:**

Strengths:

1. Innovative resolution technique: Hybrid clustering (S-BERT + k-means) followed by LLM-powered canonicalization effectively reduces sparsity, a critical bottleneck in KG extraction.

2. Benchmark contribution: MINE fills a methodological gap by directly evaluating KG fidelity to source text, avoiding conflated downstream-task metrics.

3. Rigorous evaluation: Comprehensive comparisons against GraphRAG/OpenIE across metrics (information retention, sparsity, scalability) with statistical significance reporting.

Weaknesses:

1.Computational cost: Heavy reliance on Gemini 2.0 Flash and iterative clustering may limit web-scale deployment (acknowledged but not quantified).

2. Under/over-clustering issues noted without analysis of failure modes (e.g., handling complex entities like "MRI vs. CT scan").

---

> ### Author Rebuttal · Authors · 2025-07-31
>
> Thank you for your diligent assessment of our contributions.  We address the weaknesses and limitations that you identified below.
>
>
> **Weaknesses**
>
>
> >Could the resolution stage incorporate domain-specific constraints (e.g., medical ontologies) to reduce under-clustering of specialized terms?
>
> Thanks for asking!
>
> We conducted an experiment that found that while specifying an ontology allows the extractor to deterministically adhere to the ontology, it led to a less thorough extraction. Specifically, we ran the extractor over a family tree description and found that only 14/39 were identified with the family ontology whereas 39/39 related people were identified without one. This is currently the subject of a follow-up work, in which we are developing a version of KGGen that can ingest an ontology to better guide its extraction.
>
> >What is the throughput (tokens/second) of KGGen on 1M-token corpora, and how does cost scale vs. GraphRAG?
>
>
> We currently do not have accurate cost numbers for GraphRAG aside from time required for extraction since GraphRAG makes LLM calls internally. However, we can provide this data for KGGen with Gemini. On a 1M-token corpus, here is the breakdown for cost and time, along with the throughput:
>
>
> | Step | Prompt Tokens | Completion Tokens | Total Tokens | Time (s) | Tokens/s | Cost ($) |
> |----------------------------------------------------------|---------------|-------------------|--------------|----------|----------|----------|
> | KG Extraction (Steps 1–2) | 1592718 | 626481 | 2219199 | 272.65 | 8139.4 | 0.462726 |
> | Entity/Edge Resolution (Step 3)   | 2925846 | 222790 | 3148636 | 278.54 | 11304.2 | 0.381701 |
> | KGGen TOTAL | 4518564 | 849271 | 5367835 | 551.19 | 9738.7 | 0.844427 |
> | GraphRAG extraction |- |- |- | 2318.90|-|-|
>
> >In MINE-1, why use an LLM for fact verification instead of human annotation? Could model biases affect scores?
>
>
> This is a reasonable question.  Scoring by hand is extremely labor intensive.  However, we scored 60 questions by hand and compared our judgements to those of GPT-4o.  We found a 90.2% agreement rate and a 0.80 correlation.  Therefore, LLM-as a judge is fairly accurate for this application.
>
>
>
>
>
>
>
>
> >The authors claims KGGen has already enjoyed widespread community adoption. KGGen has received over 450 Github stars and has been downloaded as a python module over 11,000 times since its release in January. Thus, KGGen has already made an impact in the community that studies knowledge graphs. Does this violate the double-blind policy?
>
>
> We deliberately redacted the package name.  There are many python modules that have been downloaded 11,000 times in the last several months, so this is not a unique identifier of KGGen.
>
>
> **Limitations**
>
>
> >Generalizability: Evaluation primarily on Wikipedia-style articles; performance on conversational or technical text is unknown.
>
>
> We ran KGGen on some fictional works that contain dialogue as well.  However, it is much more difficult to evaluate performance on non-factual genres.  Moreover, the primary use-cases for knowledge-graphs involve factual text and databases.  MINE-1 and MINE-2 incorporate a broad array of texts to help assess KGGen holistically.  Mine 2 is evaluated on the public WikiQA dataset, which contains 20,400 questions based on 1,995 wikipedia articles, and the details of MINE-1 can be found below.
>
>
> **MINE1:**
> Number of articles: 105
> Mean Length: 592 Words
> Std. Length: 85 Words
> Min Length: 440 Words
> Max Length: 976 Words
> Topic Breakdown:
> - Arts, Culture & Society: 24 essays
> - Science: 27 essays
> - Technology: 19 essays
> - Psychology/Human Experience: 18 essays
> - History & Civilization: 17 essays
> >Black-box dependencies: Entity merging relies on Gemini without error analysis (e.g., confusion rates for ambiguous entities).
>
>
> This is a fair criticism, which is common to many LLM-applications today.  We are committed to improving error analysis, and we believe that we will have more insight into the necessary steps towards this goal as KGGen enjoys more widespread adoption.

---

> > ### Comment · Reviewer_S7Xx · 2025-08-04
> >
> > Thanks for the reply, I have read and decided to keep my positive score.

---

### Official Review · Reviewer_fE3b · 2025-06-30

**Clarity:** 3
**Significance:** 3
**Originality:** 3
**Rating:** 4
**Confidence:** 3

**Summary:**

This paper introduces KGGen, a generator for extracting knowledge graphs from unstructured plain text. KGGen uses language
models to extract high-quality graphs from plain text with a novel entity resolution approach. To evaluate their generator, the authors propose a new two-part benchmark, MINE (Measure of Information in Nodes and Edges). The results show that KGGen outperforms baselines like GraphRAG and OpenIE on MINE-1 and achieves comparable performance to GraphRAG on MINE-2, The work is also presented as an open-source Python package to encourage adoption.

**Questions:**

**Suggestions/Questions**:

1. What is the evaluation accuracy of the LLM-based scoring in the benchmark MINE-1?
2. Providing a detailed comparision of efficiency and cost between different methods.
3. Paying attention to the paper writting (repeated sections).
4. Providing more detailed analysis to key modules of the proposed method.
5. How to process knowledge conflict when creating KGs?

I'm willing to raise my score if these suggestions/questions are solved.

**Ethical Concerns:**

["NO or VERY MINOR ethics concerns only"]

**Final Justification:**

My concerns have been solved.

**Limitations:**

Yes

**Paper Formatting Concerns:**

There are two "Limitations and Future Work" sections (Section 6 and Section 9) and two "NeurIPS Paper Checklist" in the paper.

**Quality:**

3

**Strengths And Weaknesses:**

**Strengths:**
1. The paper is generally well-written and clearly structured.
2. The authors have not only proposed the KGGen method for the important text-to-knowledge-graph task, but also implemented it as an useful open-source Python package.
3. The proposal of benchmark MINE is a valuable conceptual contribution.

**Weaknesses:**
1. The MINE-1 scoring mechanism relies on an LLM-as-a-Judge to determine if a fact can be inferred from a KG subgraph. The reliability, bias, and accuracy of this judge are unknown and uncalibrated.
2. The paper's ultimate goal is to create more useful KGs. However, on the MINE-2 RAG task, KGGen only performs "comparably" to GraphRAG.
3. The paper is missing several crucial analyses. There are no ablation studies to isolate the impact of the key modules of the KGGen. Additionally, there is no discussion or comparison of efficiency or cost (e.g., inference time, LLM API tokens/cost), which are critical factors for a system designed to operate on large text corpora.
4. The manuscript contains format errors. There are two "Limitations and Future Work" sections (Section 6 and Section 9) and two "NeurIPS Paper Checklist".
5. While the KGGen system is a well-designed piece of engineering, its core components are largely combinations of existing techniques.

---

> ### Author Rebuttal · Authors · 2025-07-31
>
> Thanks for your helpful suggestions for improvement.  We address your concerns below, and we hope that you'll reconsider your score.
>
>
> **Weaknesses and Questions:**
>
>
> >The paper's ultimate goal is to create more useful KGs. However, on the MINE-2 RAG task, KGGen only performs "comparably" to GraphRAG.
>
>
> Although GraphRAG performed similarly on retrieval benchmarks, it does not conform to the basic tenet of knowledge graphs: a knowledge graph consists of subject-predicate-object triples, where the subject and object are nodes and the predicate is an edge.  We chose 100 node-triple-node pairs from each of OpenIE, GraphRAG, and KGGen and manually evaluated the percentage of these triples that conform to the definition of a knowledge graph.  The results were as follows:
>
>
> KGGen: 98/100
> GraphRAG: 0/100
> OpenIE: 55/100
>
>
> Despite GraphRAG’s comparable performance, it doesn’t extract something that closely resembles a knowledge graph, which is another major strength of KGGen over GraphRAG.  As discussed in the paper, KGGen exhibits better scaling properties than GraphRAG as well.
>
>
> > What is the evaluation accuracy of the LLM-based scoring in the benchmark MINE-1?
>
>
> This is a reasonable question.  Scoring by hand is extremely labor intensive.  However, we scored 60 questions by hand and compared our judgements to those of GPT-4o.  We found a 90.2% agreement rate and a 0.80 correlation.  Therefore, LLM-as a judge is fairly accurate for this application.
>
>
> > Providing a detailed comparison of efficiency and cost between different methods.
> AND
> >Providing more detailed analysis to key modules of the proposed method.
>
>
> This is a fair point. While OpenIE is not LLM-based and thus cost comparison is not applicable, we can provide some insight into how costs and graph characteristics scale for KGGen as well as GraphRAG. To provide some variety in terms of genre, we extract a KG using KGGen with Gemini from the novel __Name of the Wind__, by Patrick Rothfuss.  As one can see, the number of relations grows sub-linearly, and the entity de-duplication ratio improves with scale, showing the effect of the clustering algorithm.
>
>
> | Corpus Size (chars) | Pre Entities | Post Entities | Entity Dedup Ratio | Pre Relations | Post Relations | Relation Dedup Ratio | Pre Edges | Post Edges | Edge Dedup Ratio |
> |---------------------|--------------|---------------|-------------------|---------------|----------------|---------------------|-----------|------------|------------------|
> | 100                 | 1            | 1             | 1.000000          | 1             | 1              | 1.000000            | 1         | 1          | 1.000000         |
> | 1000                | 20           | 18            | 0.900000          | 12            | 12             | 1.000000            | 9         | 9          | 1.000000         |
> | 10000               | 90           | 78            | 0.866667          | 78            | 75             | 0.961538            | 62        | 57         | 0.919355         |
> | 100000              | 727          | 604           | 0.830812          | 926           | 924            | 0.997840            | 498       | 424        | 0.851406         |
> | 1000000             | 4602         | 3573          | 0.776402          | 8137          | 8094           | 0.994715            | 3180      | 2448       | 0.769811         |
>
>
> The costs for each phase of KGGen on the entire novel can be found in the table below. We hope that the reviewer will agree that these costs are quite reasonable given the corpus size.
>
>
> | Step | Prompt Tokens | Completion Tokens | Total Tokens | Time (s) | Tokens/s | Cost ($) |
> |------|---------------|-------------------|--------------|----------|----------|----------|
> | KG Extraction (Steps 1–2) | 1592718 | 626481 | 2219199 | 272.65 | 8139.4 | 0.462726 |
> | Entity/Edge Resolution (Step 3) | 2925846 | 222790 | 3148636 | 278.54 | 11304.2 | 0.381701 |
> | TOTAL | 4518564 | 849271 | 5367835 | 551.19 | 9738.7 | 0.844427 |
>
> For comparison, we also evaluated GraphRAG using the same corpus and model. Note that we do not have accurate token usage and cost data for GraphRAG since it makes LLM calls internally.
>
> | Corpus Size (chars) | Entities | Relations | Edge Types | Time (s) |
> |---------------------|----------|-----------|------------|----------|
> | 100                 | 2        | 1         | 1          | 1.89     |
> | 1000                | 4        | 3         | 3          | 3.01     |
> | 10000               | 16       | 20        | 20         | 29.71    |
> | 100000              | 80       | 100       | 99         | 205.12   |
> | 1000000             | 514      | 981       | 966        | 2079.17  |
>
> >How to process knowledge conflict when creating KGs?
>
>
> Currently we don’t have a general solution to this problem.  Resolving knowledge conflict would likely require post-processing whose specifics vary based on the corpus.
>
>
> > Paying attention to the paper writing (repeated sections).
>
>
> We apologize for this mistake.  We forgot to reformat the paper until very near the deadline, and we were rushing to adapt the paper for the NeurIPS template: mistakes were made.  While we are not allowed to upload a new manuscript during the rebuttal period, please be assured that if our work is accepted, we will ensure that the writing is cleaner and limitations only appear once.

---

### Official Review · Reviewer_6cS2 · 2025-07-01

**Clarity:** 4
**Significance:** 3
**Originality:** 4
**Rating:** 5
**Confidence:** 4

**Summary:**

The authors in this paper introduce KGGen, a novel system for extracting KGs from plain text using LMs and entity resolution – identifying nodes that refer to the same underlying entities and consolidating similar/equivalent edges. The authors propose that KGGen addresses graph sparsity with this approach, unlike OpenIE and GraphRAG by utilizing LLM-guided clustering and resolution. They also propose MINE-1 for information retention and MINE-2 for RAG-style retrieval performance for KG evaluation. KGGen is an open-source package available on Github.

**Questions:**

Q1: Does KGGen’s core components generalize better across other LLMs (Claude, GPT)?

Q2: What are some challenges observed with building KG from text that is domain-specific? Has this been explored by the authors? Domains like medical, financial can be quite challenging to build a KG on. This can also be thought of as a limitation since building a KG requires domain knowledge and often times the context human-evaluators/annotators carry.

Q3: What happens if the clusterings are incorrect or inaccurate?

Q4: Does KGGen take into account ontologies or ontology-aware extractions?

**Ethical Concerns:**

["NO or VERY MINOR ethics concerns only"]

**Final Justification:**

I will maintain my score as the proposed work because of the impact and the novelty in the domain.

**Limitations:**

Aside from the limitation addresses in Q2, I am interested in knowing from the authors if they have observed any biases in the LLMs and if they observed this compounding during the growth of the KGs.

**Paper Formatting Concerns:**

No concerns.

**Quality:**

3

**Strengths And Weaknesses:**

S1: The paper is well-written and is clear to follow.

S2: KGGen combines S-BERT embeddings with k-means clustering for entity and edge resolution which is quite significant to improve the coherence of the KG.

S3: The system also uses multi-stage approach for LLM-based KG extraction and enforce strong constraints on the model through prompting. This primarily aids in preventing incorrect entity groupings which is a strong step to strengthen the objective of the framework.

S4: The benchmarks proposed (MINE-1 and MINE-2) is a novel evaluation approach.


W1: Both the evaluation approaches rely on LLMs which introduces the concern of hallucinations and bias.

W2: While the work states the corpora size, it is the first occurrence of this and there is missing context on the corpora, its contents and how KGGen generalizes with the growth of the corpus/corpora.

---

> ### Author Rebuttal · Authors · 2025-07-31
>
> REBUTTAL:
>
>
> Thank you for the positive view of our work.  We are glad that you appreciated the advantages of KGGen over past methods!
>
>
> We hope that we can address some of your remaining concerns below:
>
>
> > W1: Both the evaluation approaches rely on LLMs which introduces the concern of hallucinations and bias.
>
> That is a fair point.  However, potential bias is a drawback of __any__ method that uses LLMs or even NLP, so it is not a unique weakness of KGGen.
>
> > W2: While the work states the corpora size, it is the first occurrence of this and there is missing context on the corpora, its contents and how KGGen generalizes with the growth of the corpus/corpora.
>
> Thanks for pointing this out.  MINE-2 was evaluated on the public WikiQA dataset, which consists of 20,400 questions about 1,995 Wikipedia articles.  The details of this dataset can be found publicly.  MINE-1 was evaluated on 105 synthetically generated articles.  We provide some general statistics about these articles and a breakdown of their topics below:
> Mean Length: 592 Words
> Std. Length: 85 Words
> Min Length: 440 Words
> Max Length: 976 Words
> Topic Breakdown:
> - Arts, Culture & Society: 24 essays
> - Science: 27 essays
> - Technology: 19 essays
> - Psychology/Human Experience: 18 essays
> - History & Civilization: 17 essays
>
>
>
>
> > Q1: Does KGGen’s core components generalize better across other LLMs (Claude, GPT)?
>
>
> To answer this question, we measured the performance of KGGen on MINE-1 using 2 other models:
> | Model | MINE-1 Score |
> |-------|---------------|
> | Sonnet 4 | 73% |
> | GPT-4o | 66% |
> | Gemini | 44% |
>
>
> As you can see, the KGGen generalizes well across many different foundation models.
>
>
> > Q2: What are some challenges observed with building KG from text that is domain-specific? Has this been explored by the authors? Domains like medical, financial can be quite challenging to build a KG on. This can also be thought of as a limitation since building a KG requires domain knowledge and often times the context human-evaluators/annotators carry.
>
>
> MINE-2 aims to address this question by testing KGGen on factual articles that span many different topics.  KGGen performed well on MINE-2, demonstrating its capability across domains.  We agree that sometimes human domain-experts will outperform LLMs on their areas of expertise, but this is an inherent challenge in building a KG-extractor that generalizes to arbitrary text.  Unfortunately, there are not enough domain experts to construct large-scale and comprehensive KGs, leaving the task up to methods like KGGen.
>
>
> > Q3: What happens if the clusterings are incorrect or inaccurate?
>
> During the clustering process, sub-nodes of each cluster are stored.  Therefore, if one suspects that a cluster contains an inappropriate member, one can easily remove and correct the error.
>
> > Q4: Does KGGen take into account ontologies or ontology-aware extractions?
>
> Thanks for asking this question!
>
> We conducted a preliminary experiment that found that specifying an ontology allows the extractor to deterministically adhere to the ontology, but led to a less thorough extraction. Specifically, we ran the extractor over a family tree description and found that only 14/39 were identified with the family ontology whereas 39/39 related people were identified without one.  For this reason, a version of KGGen that allows for pre-specified ontologies is currently the subject of a follow-up work.

---

> > ### Comment · Reviewer_6cS2 · 2025-08-01
> >
> > Thank you authors for addressing my concerns and questions. I am slightly concerned about KGGen's MINE-1 score on Gemini, would the authors be aware of what is contributing to this score? Additionally, the ontology leading to a less thorough extraction can point to two things, (1) the ontology would need to be refined; (2) the knowledge is too "noisy" or too "much" to adhere to the ontology. While this seems to be out of scope for this work, it is worthwhile to consider the quantity of the knowledge for future use cases for this proposed work to include ontology-aware extractions.
> > In conclusion, the responses are sufficient and I will maintain my score of 5: Accept because the authors' proposed work and impact is novel and required (respectively), in the domain.

---

### Official Review · Reviewer_vhqp · 2025-07-03

**Clarity:** 2
**Significance:** 3
**Originality:** 2
**Rating:** 4
**Confidence:** 5

**Summary:**

The paper introduces a software package for extracting knowledge graphs directly from text using Large Language Models (LLMs). Specifically, Gemini 2.0 Flash is used to extract entities (nodes) and relationships (edges) between nodes. Subsequently, a clustering approach is used to perform entity resolution and thus reduce the sparsity of the graph. To evaluate the proposed approach, the authors also introduce two datasets. Experimental results show that the proposed approach has performance comparable to that of Microsoft’s
GraphRAG, while being more scalable.

**Questions:**

Why could't you evaluate the proposed approach on existing triple datasets that are manually extracted?

**Ethical Concerns:**

["NO or VERY MINOR ethics concerns only"]

**Final Justification:**

The authors responded to my questions and concerns. However, I still think the evaluation of the approach, in its current form, may not be very reliable. Nevertheless, I increased my original score.

**Limitations:**

The authors discuss some limitations.

**Paper Formatting Concerns:**

No concerns

**Quality:**

2

**Strengths And Weaknesses:**

The authors focus on the important task of leveraging LLMs to extract knowledge graphs from plain text.

Two novel datasets for evaluating approaches for extracting knowledge graphs are introduced.

The approach proposed is very basic and not well-explained.

The datasets introduced are not well described. Some statistics about the datasets would be useful.

The experimental evaluation is done by verifying facts. It would also be good to evaluate the actual triples extracted. In fact, it is not clear why the authors can't evaluate this approach on existing knowledge graphs.

The results of the proposed approach are not better than those of one of the baselines, despite the improved scalability of the proposed approach.

The paper is not very well written. The Limitations section is included in two places.

---

> ### Author Rebuttal · Authors · 2025-07-31
>
> Thank you for taking the time to review our paper.  While we appreciate and will integrate your suggestions for improvement, we disagree with your overall characterization of our paper.  We address the identified weaknesses and questions below, and we hope that you'll reconsider your score.
>
>
> >The datasets introduced are not well described. Some statistics about the datasets would be useful.
>
>
>  We are not exactly sure what statistics would be interesting to readers.  To describe our datasets better:
>
>
> **MINE1:**
> Number of articles: 105
> Mean Length: 592 Words
> Std. Length: 85 Words
> Min Length: 440 Words
> Max Length: 976 Words
> Topic Breakdown:
> - Arts, Culture & Society: 24 essays
> - Science: 27 essays
> - Technology: 19 essays
> - Psychology/Human Experience: 18 essays
> - History & Civilization: 17 essays
>
>
> **Mine 2**:
> Mine 2 is evaluated on the public WikiQA dataset, which contains 20,400 questions based on 1,995 wikipedia articles.  We are happy to provide more specific details about this dataset from Microsoft if the reviewer is interested.
>
>
>
>
> >The experimental evaluation is done by verifying facts. It would also be good to evaluate the actual triples extracted. In fact, it is not clear why the authors can't evaluate this approach on existing knowledge graphs.
>
>
> We can’t evaluate this approach on existing knowledge graphs because there is no good metric for whether two knowledge graphs are similar.  As you point out in your review, there are no knowledge-graph foundation models due to lack of data, which leaves embedding similarity out of the question.  While there exist methods of graph embeddings, these only compare graph structure without taking node-level data into account.  Moreover, human-labeled knowledge graphs are not infallible: it is well-known that even the largest human-labeled knowledge graphs, such as WikiQA, have millions of missing nodes and relations.  Fact-verification is an efficient and objective way to measure a KG-extractor’s effectiveness.  We also note that past works, like GraphRAG and OpenIE did not perform verification via triples either.  In fact, our evaluations are already far more extensive than past works in the field, which generally verify based on metrics like information-compression.
>
>
>
>
> > The results of the proposed approach are not better than those of one of the baselines, despite the improved scalability of the proposed approach.
>
>
> We first note that improved scalability is very important for a method to be practical, and should not be downplayed.
>
>
> Although GraphRAG performed similarly on retrieval benchmarks, it does not conform to the basic tenet of knowledge graphs: a knowledge graph consists of subject-predicate-object triples, where the subject and object are nodes and the predicate is an edge.  We chose 100 node-triple-node pairs from each of OpenIE, GraphRAG, and KGGen and manually evaluated the percentage of these triples that conform to the definition of a knowledge graph.  The results were as follows:
>
>
> | Method | Fraction Valid |
> |--------|-------|
> | KGGen | 98/100 |
> | GraphRAG | 0/100 |
> | OpenIE | 55/100 |
>
> Despite GraphRAG’s comparable performance, it doesn’t extract something that closely resembles a knowledge graph, which is another major strength of KGGen over GraphRAG.
>
>
> >The paper is not very well written. The Limitations section is included in two places.
>
>
> We apologize for this mistake.  We forgot to reformat the paper until very near the deadline, and we were rushing to adapt the paper for the NeurIPS template: mistakes were made.  Please be assured that if our work is accepted, we will ensure that the writing is cleaner and limitations only appear once.
> >Why couldn't you evaluate the proposed approach on existing triple datasets that are manually extracted?
>
>
> Manually extracted KG’s are not unique, as two different humans will likely extract different KGs from the same text.  Even if there were agreement on the “correct” KG for a given text, it is very difficult to measure similarity between two knowledge graphs given that there are no foundation models to provide embeddings.

---

> > ### Comment · Reviewer_vhqp · 2025-08-06
> >
> > Thank you for your response. I appreciate the additional clarifications. I will update my score. But I still have some questions and concerns. You are defining a knowledge graph as a set of triples. There are several existing knowledge graphs (a.k.a., sets of triples) in the literature, for example,  SemEval-2010 Task 8, which is manually annotated. Can you extract triples from the sentences in SemEval-2010 Task 8 and compare the resulting set of triples to the ones from SemEval-2010 Task 8? I understand that a set of triples is not unique, but human agreement could be used to judge what you might expect to get in the best case. Error analysis could point to cases where your approach does well and where it fails.  In its current form, the evaluation does not seem to be very reliable.

---

> > > ### Author Response · Authors · 2025-08-06
> > >
> > > Thank you for this valuable feedback. You raise a valid point about evaluating against human-annotated triples. We have conducted exactly the evaluation you suggested using the SemEval-2010 Task 8 dataset.
> > >
> > > Evaluation Methodology:
> > >
> > > 1. Randomly selected 100 sentences from the SemEval-2010 Task 8 dataset, each containing 2 manually labeled target entities and removed the entity markup tags (e.g., < e1 >, < /e1 >) from sentences before passing them to KGGen to ensure unbiased extraction.
> > >
> > > 2. Applied KGGen to each cleaned sentence to extract entities and relationships.
> > > For each sentence, manually determined whether both labeled target entities appeared in KGGen's extracted entities. Entities that were more specific than the human annotation but referred to the same object in the sentence were not penalized.
> > >
> > > Example:
> > >
> > >     Sentence: "The ambitious Eurasia exhibition arose from an idea by Achille Bonito Oliva."
> > >     Target Entities: 'exhibition', 'idea'
> > >     KGGen Entities: ['Eurasia exhibition', 'idea', 'Achille Bonito Oliva']
> > >     Result: Success (both target entities captured, with KGGen providing more specific entity identification)
> > >
> > > Our evaluation focused on entity extraction accuracy since the dataset's relation labels consist of only 5 broad categorical types rather than specific semantic relations, making meaningful relationship comparison unfeasible. In 96/100 cases, KGGen's extracted entities successfully contained both human-annotated target entities.
> > >
> > > We found that KGGen consistently extracts more detailed entity descriptions (e.g., "rock band" vs. "band") compared to human annotations. In longer sentences, KGGen also identifies additional entities beyond the annotated pair, which appeared appropriate given that such sentences contained information sufficient for multiple semantic relationships.

---

> > > > ### Comment · Reviewer_vhqp · 2025-08-09
> > > >
> > > > Thank you for adding this experiment. I updated my score.

---

### Decision · Program_Chairs · 2025-09-17

**Decision:**

Accept (poster)

**Comment:**

This paper introduces KGGen, a language-model-based framework for extracting knowledge graphs from plain text with entity/relation resolution, accompanied by the new MINE benchmarks for evaluating information retention and downstream retrieval performance. Reviewers appreciated the novelty, clarity, and practical impact of the work, highlighting its open-source release, strong performance against GraphRAG and OpenIE, and meaningful contributions through benchmark design. While some concerns were raised regarding evaluation reliability (LLM-as-a-judge), dataset descriptions, and missing ablations, the authors provided additional experiments and clarifications during rebuttal that addressed most issues. Overall, the paper makes a solid methodological and resource contribution to the growing area of text-to-KG extraction and is recommended for acceptance.